# Effect of thermal ballast loading on temperature stability of domestic refrigerators used for vaccine storage

Michal Chojnacky[1]*, Alexandra L. Rodriguez[1,2]¤

1 Physical Measurement Laboratory, National Institute of Standards and Technology, Gaithersburg, Maryland, United States of America, 2 Department of Computer Science, University of Maryland, College Park, Maryland, United States of America

¤ Current address: Independent Researcher, Bethesda, Maryland, United States of America
* michalc@nist.gov

**Data Availability Statement:** All relevant data are within the manuscript and its Supporting Information files.

**Funding:** Research reported in this publication was supported by the Centers for Disease Control and

## Abstract

Vaccine temperature control failures represent a significant public and private healthcare cost. Vaccines damaged by excessive heat or freezing lose their effectiveness, putting public health at risk. Some vaccine administration programs recommend placing water bottles inside domestic refrigerators used for vaccine storage as a thermal ballast, to mitigate temperature excursion risks. However, the effect of variable thermal ballast loading on refrigerator performance has not been thoroughly quantified or documented, and generalized programmatic recommendations are subject to end-user interpretation. Here we show that a thermal ballast load comprising ten to fifteen percent of the total refrigerator storage volume provides a measurable effect on domestic refrigerator temperature stability during power outage events, maintaining vaccine temperatures between 2 ˚C and 8 ˚C for 4 to 6 hours without power. Thermal ballast usage does not reliably reduce the frequency or severity of temperature excursions caused by repeated door opening, accidental "door left open" events, or refrigerator defrost cycle activation. Use of a moderate thermal ballast load is a practical strategy for mitigating temperature excursions risks in areas with frequent or protracted power outages, but the practice has limited benefit in other adverse scenarios. Empowering providers to make informed decisions about the use of thermal ballast materials supports better, safer vaccine management.

## Introduction

Effective temperature-controlled storage is vital to preserving vaccine potency. In the United States, most vaccines must be maintained at temperatures between 2 ˚C and 8 ˚C from the point of product manufacture until delivery to a patient [1]. Prolonged and extreme excursions from this range can compromise the efficacy of the vaccine. Most of these vaccines are freeze-sensitive, as exposure to temperatures below 0 ˚C induces morphological changes in the antigens and/or adjuvants, leading to irrevocable potency losses [2–4].

Prevention (CDC) [Interagency Agreement 13FED1310110], and the authors' institution, the National Institute of Standards and Technology (NIST). The content is solely the responsibility of the authors and does not necessarily represent the official views of the CDC or NIST. CDC Vaccines for Children website: https://www.cdc.gov/vaccines/programs/vfc/ The CDC identified a research question applicable to the administration of its Vaccines for Children program: is there a minimum quantity of thermal ballast materials that provides improved temperature stability of vaccines stored in refrigerators? This study was conceptualized to answer that question and to provide technical guidance on implementation of thermal ballast loading at the vaccine provider level. Beyond this, the funders had no role in the study design; the collection, analysis and interpretation of data; in the writing of the report; or in the decision to submit the article for publication.

**Competing interests:** I have read the journal's policy and the authors of this manuscript have the following competing interests: This study was partially funded by an Interagency Agreement with Centers for Disease Control and Prevention (CDC) and the authors' institution. As part of this agreement, MC provides technical guidance and research support to the CDC's Vaccines for Children program. Additionally, MC is a member of the NSF Joint Committee for Vaccine Storage, a consensus body responsible for creating and maintaining an ANSI standard for purpose-built vaccine storage equipment. This does not alter our adherence to PLOS ONE policies on sharing data and materials.

Present-day Centers for Disease Control (CDC) guidance permits the use of domestic refrigerator units for vaccine storage [5]. While these units are designed and marketed for home food storage, their low cost, availability, and large storage capacity make them attractive purchases for vaccine providers. Domestic refrigerator units may be used to safely maintain stored vaccines when paired with proper user training and temperature monitoring techniques [6,7]. However, field studies in the United States, Australia, and Tunisia documented repeated exposures to temperatures outside the permissible range in domestic refrigerators used for vaccine storage, and the widespread use of domestic refrigerators is frequently cited as a cause of vaccine wastage [8–12].

Prior studies indicated that placing added thermal mass in the form of water bottles inside a vaccine storage refrigerator may improve temperature stability and prolong viable vaccine storage temperatures during adverse events such as power outages, defrost cycle activation, and routine door opening [6,7]. Current CDC guidelines recommend storing water bottles inside domestic refrigerators to prevent vaccine storage in unsuitable regions (like the refrigerator door), and to act as a "thermal ballast," stabilizing vaccine temperatures and minimizing excursions [5]. However, the quantity of thermal ballast materials needed to improve the stability of stored vaccine temperatures has not been thoroughly studied or documented, so existing recommendations are subject to interpretation by users and immunization program administrators.

In this study, we examined the effect of thermal ballast loads on the temperature stability of vaccine stored in two domestic refrigerators under normal operation and during adverse events. Our primary objective was to determine the minimum quantity of thermal ballast materials required to reduce the incidence or severity of temperature excursions, as compared to the same units used without thermal ballast materials.

Each refrigerator was tested using four ballast loads, in which 25%, 17%, 8%, or 0% of the total refrigerator storage volume was filled with water bottles. Stored vaccine temperatures were monitored during closed-door, steady-state operation, defrost cycle activation, repeated door opening, and power loss events, to assess the impact of each load percentage on the stored product temperature. In addition, different storage containers (open plastic bins, covered plastics bins, and metal trays) were included to investigate whether the temperature stability of refrigerated vaccine is correlated with storage container type.

## Materials and methods

### Refrigerator selection

Two domestic refrigerator units were included in this study: a combination refrigerator/freezer, and a standalone refrigerator. While both designs are marketed as consumer-grade, food storage refrigerators, these types of units are frequently used for vaccine storage due to their low cost and availability [11–14]. In the 2019 Vaccine Storage and Handling Toolkit [5], the CDC recommends storing vaccine in a purpose-built refrigerator designed specifically for pharmaceutical storage. However, if a purpose-built unit is not available, a domestic standalone unit is suggested as an acceptable alternative. Domestic combination refrigerator/freezer units are generally not recommended for vaccine storage due to an increased risk of accidental freezing. However, current guidance permits the use domestic combination units for refrigerated vaccine storage in the absence of recommended alternatives [5]. Since domestic refrigerators are designed for food storage, these units are typically not engineered to match the level of temperature control and stability provided by a purpose-built vaccine storage refrigerator. In fact, meta-analyses of domestic food storage refrigerator temperature studies from 1979 to

2016 indicate that household units used worldwide routinely operate outside the recommended refrigerated food storage temperature range of 0 ˚C to 8 ˚C [15,16].

Both units used in this study featured an upright, swinging door design. The combination unit had three glass storage shelves inside its refrigerated compartment, and the standalone unit had four interior wire shelves. The standalone unit was also tested with an aftermarket glass shelf installation as a separate case study, described in later sections of this article. Both units were housed in the National Institute of Standards and Technology (NIST) Gaithersburg Thermal Studies Laboratory throughout the study. Refrigerator details are summarized in Table 1.

Both refrigerators featured an analog temperature control dial, which was kept on a mid-point setting throughout testing. This corresponded to a nominal temperature set point of 4 ˚C to 5 ˚C in each refrigerator.

## Vaccine storage setup

Both refrigerators were outfitted with storage containers arranged in the center of each unit, filling approximately 50% of the available shelf space. The top shelf, floor, door, and approximately 15 cm of space adjacent to each interior side wall was kept free of stored vaccines throughout the study to allow for placement of water bottles in these areas. Each of the remaining shelves was outfitted with two closed plastic bins, one large metal tray, and two uncovered plastic bins (Fig 1). The storage containers were filled with vaccine vials and pre-filled syringes stored in their original, manufacturer-supplied cardboard packaging, along with unboxed vaccine vials placed directly in the bin or tray. Two vaccines from each container, one boxed and one unboxed, were selected for temperature monitoring (Fig 2). Vaccine and storage container positions remained fixed throughout the study. The vaccines used in this study included a mix of 0.5 mL single dose vials, 0.5 mL prefilled syringes, and 5 mL multidose vials of common U. S. childhood vaccines designated for refrigerated storage between 2 ˚C and 8 ˚C, such as influenza, diphtheria and tetanus, hepatitis A and B, pneumococcal disease, and haemophilus influenzae type b virus.

## Temperature monitoring

A total of twenty-nine Type T thermocouples were used to monitor vaccines in the two refrigerators, with twelve monitored vaccines in the combination unit, and seventeen monitored vaccines in the standalone unit. Each thermocouple sensing junction was immersed in the 0.5

**Table 1. Domestic test refrigerator attributes.**

|  | Combination Refrigerator/Freezer | Standalone Refrigerator |
|---|---|---|
| Model[a] | Kenmore 253.78179801 | Kenmore 253.60722 |
| Automatic Defrost | Yes | Yes |
| Freezer | Yes, top freezer | No |
| Refrigerated compartment capacity: | 0.40 m$^3$ (14.13 ft$^3$) | 0.47 m$^3$ (16.7 ft$^3$) |
| Shelf material (quantity) | Glass (3 shelves) | Resin-coated wire (4 shelves) |
| Electrical rating | 4.5 A / 115 V / 60 Hz | 5 A / 115 V / 60 Hz |
| Refrigerant | R134a | R134a |
| Manufacture date | November 2009 | May 2009 |

[a] Commercial equipment is identified to specify the experimental procedure adequately. Such identification is not intended to imply recommendation or endorsement by the National Institute of Standards and Technology, nor is it intended to imply that the equipment identified is necessarily the best available for the purpose.

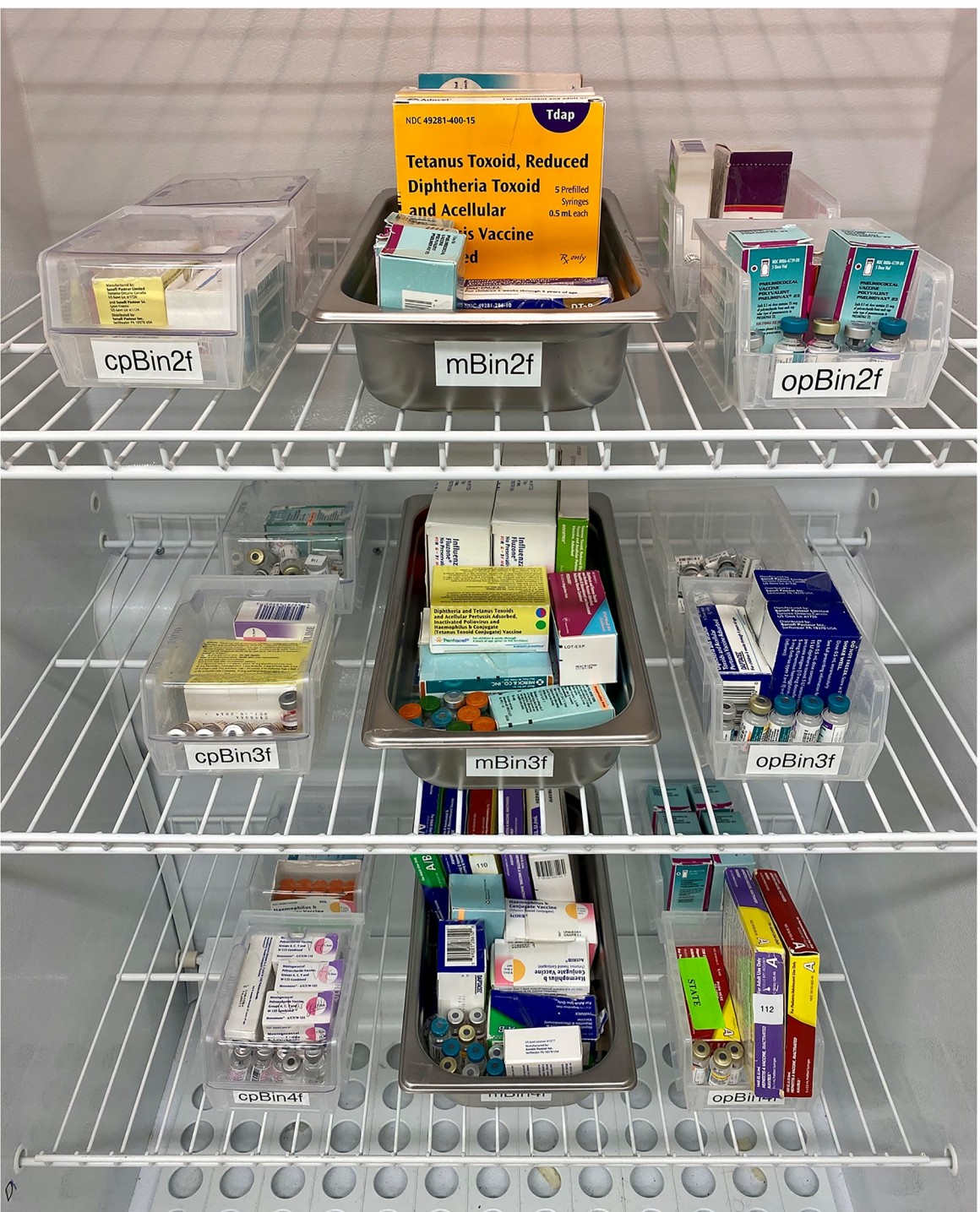

**Fig 1. Stored vaccine setup.** Each shelf was outfitted with five storage containers (left to right): two closed plastic bins, one large metal tray, and two uncovered plastic bins. The storage containers were filled with a mix of boxed and unboxed vaccines. The top shelf (not shown) was kept free of stored vaccines to allow for water bottle placement in this area.

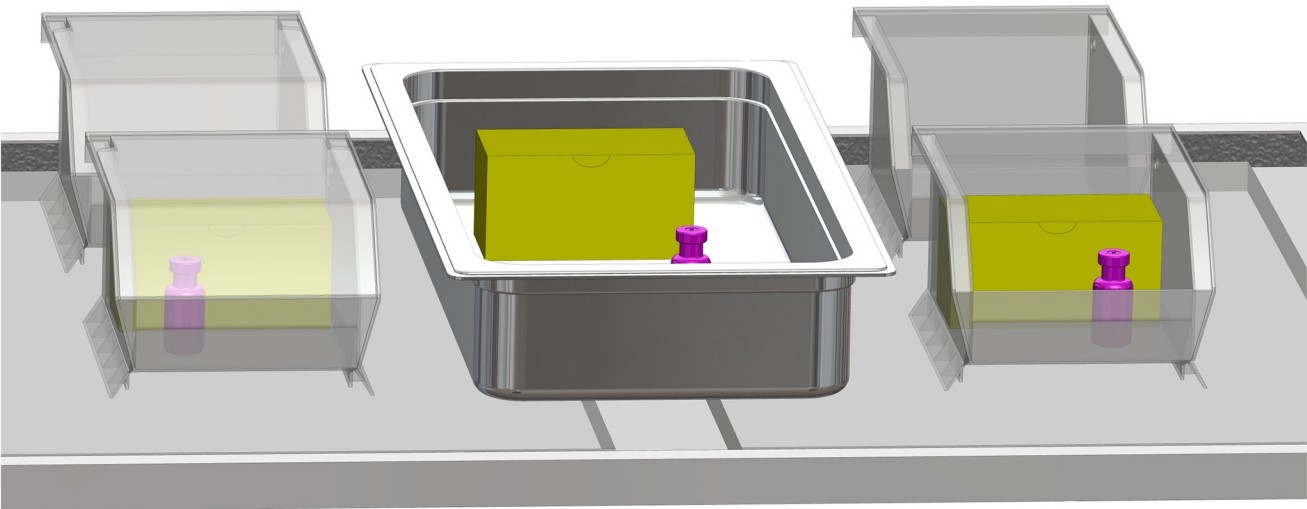

**Fig 2. Monitored vaccine vial setup.** Three boxed vaccines (shown in yellow) and three unboxed vaccine vials (shown in pink), kept in bins as shown on each shelf, were outfitted with thermocouple probes for temperature monitoring during the study. Additional, unmonitored vaccine boxes and vials, not shown in this illustration, were used to fill each bin (See Fig 1).

mL vaccine fluid volume contained in a vial or pre-filled syringe. Thermocouple reference junctions were maintained in a circulating water bath controlled to 23 ˚C ± 0.005 ˚C. Prior to their use in the study, each thermocouple was calibrated at the melting point of ice (0 ˚C) by immersing the thermocouple sensing junction in a Dewar flask containing crushed ice and distilled water, as described in NIST Technical Note 1411 [17]. The ice point measurements were used to generate a temperature offset for each thermocouple, which was applied to all measurements collected during the study. Thermocouple voltages were read via automated 6 ½ digit multimeter data acquisition units, which were set to scan and log readings to a computer every 10 seconds. The uncertainty of the temperature measurement system is shown in Table 2.

**Table 2. Temperature measurement system uncertainties[a].**

| Type A uncertainties | |
|---|---|
| | $u_i$ / ˚C |
| Thermocouple accuracy | 0.01 |
| Thermocouple repeatability | 0.05 |
| Total A | 0.05 |
| **Type B uncertainties, (rectangular distribution)** | |
| Multimeter accuracy | 0.01 |
| Multimeter resolution | 0.0005 |
| Thermocouple reproducibility | 0.09 |
| Reference junction bath stability | 0.005 |
| Total B | 0.05 |
| Total Standard Uncertainty (k = 1) | 0.07 |
| **Total Expanded Uncertainty (k = 2)** | **0.15** |

[a]Measurement uncertainties calculated according to the Guidelines for Evaluating and Expressing the Uncertainty of NIST Measurement Results [18].

## Thermal ballast

Each refrigerator was loaded and tested with varying quantities of water bottles to assess the effect of thermal ballast loading on temperature stability. Commercially available plastic drinking water bottles in the ubiquitous 500 mL (16.9 fl oz) size were used throughout the study. Larger commercially available plastic bottles and jugs were also included in some of the larger thermal ballast loads, ranging from 1000 mL (33.8 fl oz) bottles to 3950 mL (1.04 gal) jugs. The total water bottle volume used for each ballast load percentage was calculated based on the manufacturer-specified storage volume of the refrigerator under test. Water bottles were distributed in accordance with vaccine management guidelines issued by the Centers for Disease Control and Prevention (CDC), with the bulk of the bottles concentrated in the refrigerator door, on the top shelf, and floor level of the unit [5]. The relative distribution of water bottles was kept consistent for each of the 8%, 17%, and 25% ballast load configurations. In each configuration, approximately 27% of total bottle volume was in stored in the refrigerator door, 30% on the refrigerator floor, 23% on the top shelf, and the remaining 20% distributed near the walls of the intermediate shelves. Fig 3 shows this distribution applied to a 10% ballast load in the standalone unit. Additional photos of 25% and 8% thermal ballast loads are shown in S1 Fig.

## Test protocol

The test protocol was designed to probe the effect of varying thermal ballast loads on refrigerator performance during normal operating conditions and typical adverse events. "Normal operation" test modes included 1) closed door operation and 2) repeated door opening. The closed-door operation test was executed for a minimum of 24 h, to ensure capture of at least one defrost cycle event. In the repeated door opening test, the refrigerator door was held open for 30 s at 5 min intervals, over a period of 2 h, to simulate frequent vaccine retrieval during a busy clinic workday. The "adverse events" examined in this study were 3) door left open at an angle of 90˚ for 1 h, and 4) a power outage event, in which the refrigerator was unplugged and monitored for 24 h. The four-test sequence was replicated for each thermal ballast load (25%, 17%, 8%, 0%).

Both refrigerator units were tested according to this protocol. Vaccines and thermocouples were arranged in each unit prior to testing and kept fixed throughout the study. The maximum thermal ballast load (25%) was arranged in the unit prior to the first test, and temperatures were monitored for a minimum of 24 h to ensure that steady state operating conditions were achieved before testing commenced. Following completion of the four-test sequence, water bottles were removed from the unit to achieve the second-largest ballast load (17%), and again, temperatures were monitored for a minimum of 24 h to ensure equilibration before proceeding with the next test sequence. This process was repeated for the remaining ballast loads (8%, 0%).

## Shelf material case study

We also investigated possible effects of refrigerator shelf construction on vaccine storage conditions in a related case study. In this investigation, the standalone refrigerator was outfitted with vaccine-filled trays as before, and vaccine vials distributed within the storage trays were monitored with thermocouple probes to map out temperature conditions inside the refrigerator storage volume, as shown in Figs 1 and 2.

In the first set of trials, the trays were placed, as before, directly on the refrigerator's resin-coated wire shelves. In the second set of trials, glass sheets were cut to match the dimensions of the existing wire shelves, and then installed on top of the wire shelves, with two sheets placed

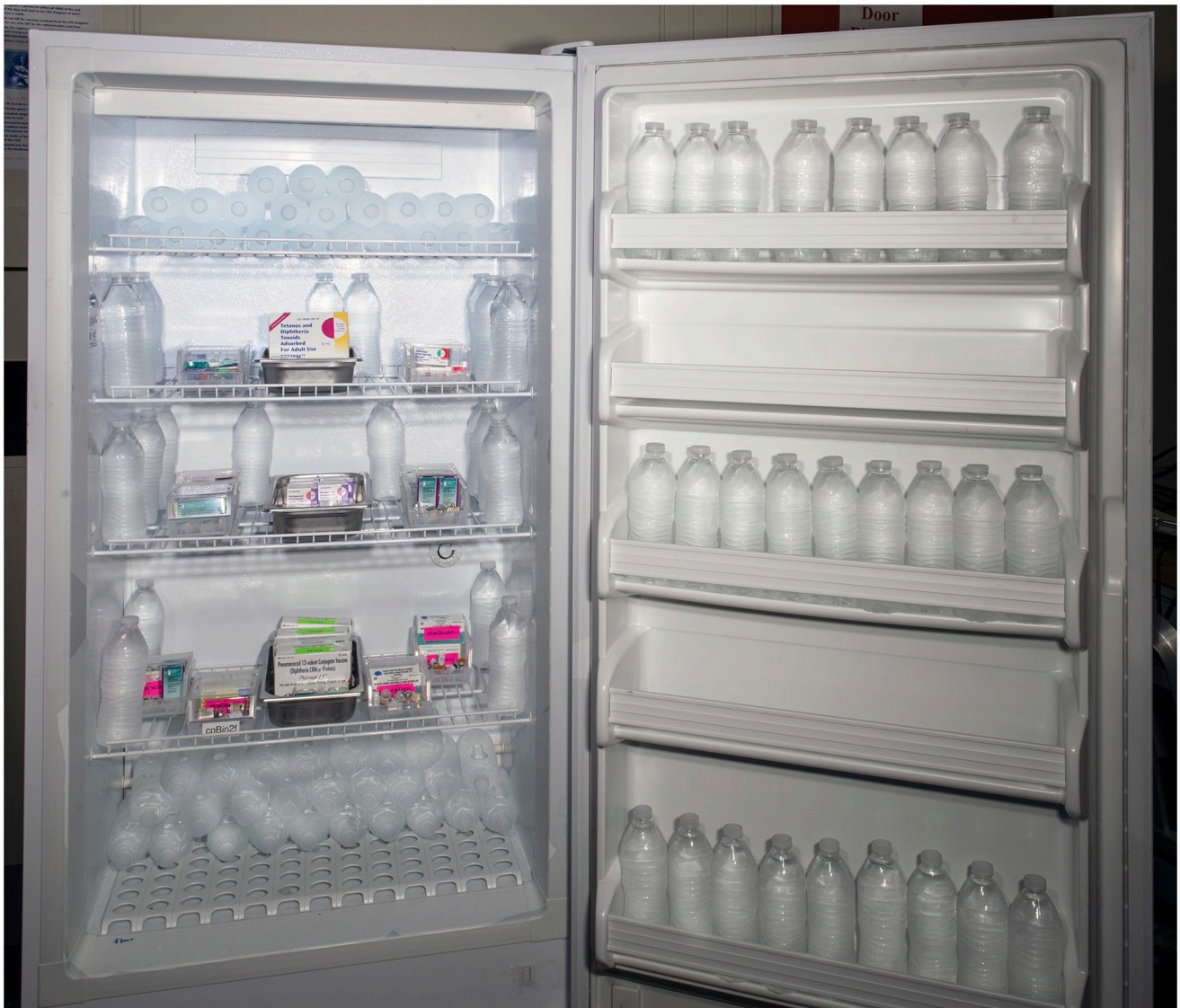

**Fig 3. Standalone refrigerator filled with a 10% thermal ballast load comprised of 500 mL (16.9 oz) plastic drinking water bottles.** Water bottles were concentrated in door, top shelf, and floor level of the unit, with a small number of bottles placed along the side walls of the unit. The relative distribution of thermal ballast (water bottles), detailed in the text, was kept fixed throughout the study. Plastic and metal bins placed in the center of the unit were used to contain a mix of boxed and unboxed vaccine vials.

side by side on each shelf. Sheets were cut from consumer grade clear replacement glass. Final dimensional specifications are listed in Table 3.

The installed glass sheets were positioned to preserve an air gap between the refrigerator walls and the edges of the shelves to avoid completely obstructing the refrigerator airflow. The

**Table 3. Glass sheet specifications.**

| Installation location | Length (cm) | Width (cm) | Thickness (cm) |
|---|---|---|---|
| Shelves 1 (top), 2, 3 | 40.5 | 30.5 | 0.238 |
| Shelf 4 (bottom) | 35.5 | 30.5 | 0.238 |

air gap between the back wall of the refrigerator and the installed shelves was approximately 1 cm, and the air gap between the side walls and the adjacent edge of each shelf was approximately 3.5 cm. After installation, vaccine storage trays were placed on the glass shelves, following the same loading pattern used in the thermal ballast trials.

Each of the setups described above was subjected to a power outage test and a repeated door opening test, utilizing the same procedures described in the Thermal Ballast Test Protocol subsection. No thermal ballast materials were placed in the unit during the shelf material case study tests.

## Evaluation methods

Each measurement trial in the described in the preceding sections yielded a time-temperature data series of twelve or seventeen temperature sensors, with data points every 10 s for periods of hours up to several days. To compare and draw meaningful conclusions from these data sets, we summarized the results from each trial according to the methods described in this section.

While temperature cycling is normal and expected in domestic refrigerators, very large fluctuations can result in temperatures outside recommended vaccine storage temperature limits. Since the recommended set point for refrigerated vaccine storage is 5 ˚C, a unit that routinely cycles through a range of 3 ˚C or more may subject stored vaccines to excursions outside 2 ˚C to 8 ˚C. As a result, the temperature range, $\Delta T = T_{max} - T_{min}$, recorded by each monitored vaccine during each trial, is taken as an indicator of thermal stability. A simple mean of the $\Delta T$ for all monitored vials is given by $\overline{\Delta T}$. We therefore define $\overline{\Delta T} > 3°C$ as the condition for a vaccine temperature excursion, and $\overline{\Delta T} < 3°C$ as the condition for proper vaccine temperature maintenance. Defining temperature excursions in this way allows us to separate our evaluation from small shifts in absolute refrigerator set point temperature.

Similarly, we define "viable vaccine storage time" as the duration of time in which vaccines were maintained within 3 ˚C of their initial storage temperature ($\overline{\Delta T} < 3°C$) during a power loss or door left open event. In this framework, a temperature increase of more than 3 ˚C is equivalent to the start of a temperature excursion, and the end of viable vaccine storage time.

In the defrost cycle and door opening trials, $\overline{\Delta T}$ is used to assess refrigerator stability at different ballast loading levels. In the door left open and power outage trials, viable vaccine storage time is used as the primary assessment parameter.

In addition, a Pearson correlation coefficient, $r$, was calculated assess the strength of the linear relationship between thermal ballast load percentage and viable vaccine storage time during power outage trials. The formula for $r$ is the covariance of the two data sets divided by the product of their standard deviations:

$$r = \frac{\sum (x - \overline{x})(t - \overline{t})}{\sqrt{\sum (x - \overline{x})^2 \sum (t - \overline{t})^2}} \tag{1}$$

Here, $x$ denotes the thermal ballast load percentage for a measurement, $t$ denotes the viable vaccine storage time for a particular measured vial in a particular trial, and $\overline{x}, \overline{t}$ are the sample means of these variables, calculated from all sensor outputs in all four trials conducted in each unit.

## Results

### Defrost cycle

Normal refrigerator operation causes periodic temperature cycling of stored vaccines, even when the door is kept closed. The magnitude and frequency of stored vaccine temperature

fluctuations necessarily varies by refrigerator model, temperature setting, stored vaccine load, and environmental operating conditions. Larger, transient fluctuations typically occur during defrost cycle activation and the subsequent temperature recovery period. Basic domestic refrigerators periodically initiate a defrost cycle that includes operation of a heater attached to the evaporator coils [18,19]. This process melts accumulated frost from the coils, and as a side effect, produces a transient rise in cabinet temperature. Once the defrost heater deactivates, the refrigerator operates the cooling cycle for an extended period to cool the unit back to its set point. During the recovery phase, the cabinet temperature may drop below the setpoint before the compressor cycles off. In both units tested in this study, a measurable, transient temperature rise associated with defrost cycle activation occurred approximately once every 24 h. An example of the defrost cycle temperature response in the standalone refrigerator used in this study is shown in S1 Fig.

The defrost cycle data are summarized as described in the Evaluation methods, and the results are displayed in Fig 4. $\overline{\Delta T} > 3°C$ indicates a condition likely to result in vaccine temperature excursions.

In the standalone unit, the $\overline{\Delta T} > 3°C$ condition occurred during one trial, in which no thermal ballast was stored inside the refrigerator. Depending on the refrigerator setpoint, a fluctuation of this magnitude could expose stored vaccine to a small temperature excursion. In this unit, filling 25% of the refrigerator storage volume with water bottles reduced defrost cycle-induced temperature fluctuations by two-thirds. A more practical 8% thermal ballast load reduced temperature cycling by one-half, such that all monitored vaccine temperatures were maintained within the 3 ˚C excursion threshold.

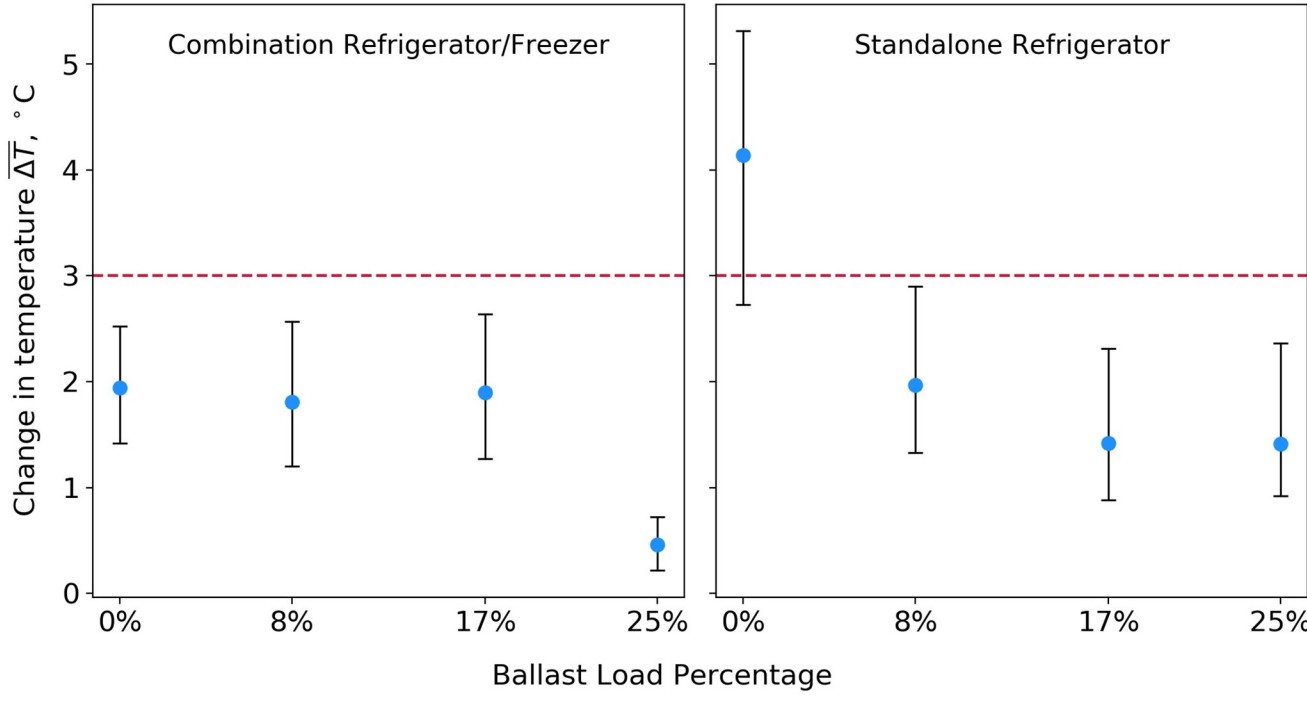

**Fig 4. Refrigerator cabinet temperature changes during defrost cycle activation at varying thermal ballast load percentages.** Round blue markers indicate the average temperature rise of all monitored vaccines during each trial, and error bars show the range of these measurements, corresponding to the within-trial variation among monitored vaccines.

In the combination unit, no $\overline{\Delta T} > 3°C$ conditions occurred during any of the defrost cycle trials. Filling 25% of the refrigerator volume with water bottles reduced temperature fluctuations by 70%. However, smaller ballast loads had little effect on the defrost cycle temperature response. As shown in Fig 4, temperature cycling was nearly identical for the 0%, 8%, and 17% loads.

## Repeated door opening

Repeated door opening in vaccine refrigerators also contributes to thermal cycling of vaccines [6,7,20]. The results of the repeated door opening trials are shown in Fig 5.

In the standalone unit, the average vaccine temperature variation exceeded 3 °C during all trials, regardless of thermal ballast load. While the temperature response varied between trials, the within-trial variation across all monitored vaccines, as indicated by error bars in Fig 5 (right), exceeded the trial-to-trial variation in all but the 8% thermal ballast trial. Within-trial variations may be attributed to thermal gradients inside the refrigerator, which increase with repeated door opening [21]. Adding thermal ballast to the standalone refrigerator does not appear to mitigate temperature instability caused by repeated door opening events, as vaccine temperature excursions occurred at all ballast loading levels tested during the repeated door opening trials.

By contrast, increased thermal ballast load appears to be correlated with improved temperature stability in a combination unit at higher ballast percentages (18%, 25%) subjected to repeated door opening. At these higher load percentages, the average temperature variation dropped below the 3 °C threshold. During the 25% thermal ballast load door opening trial, temperature fluctuations were controlled to within 2.1 °C or less across all monitored vials.

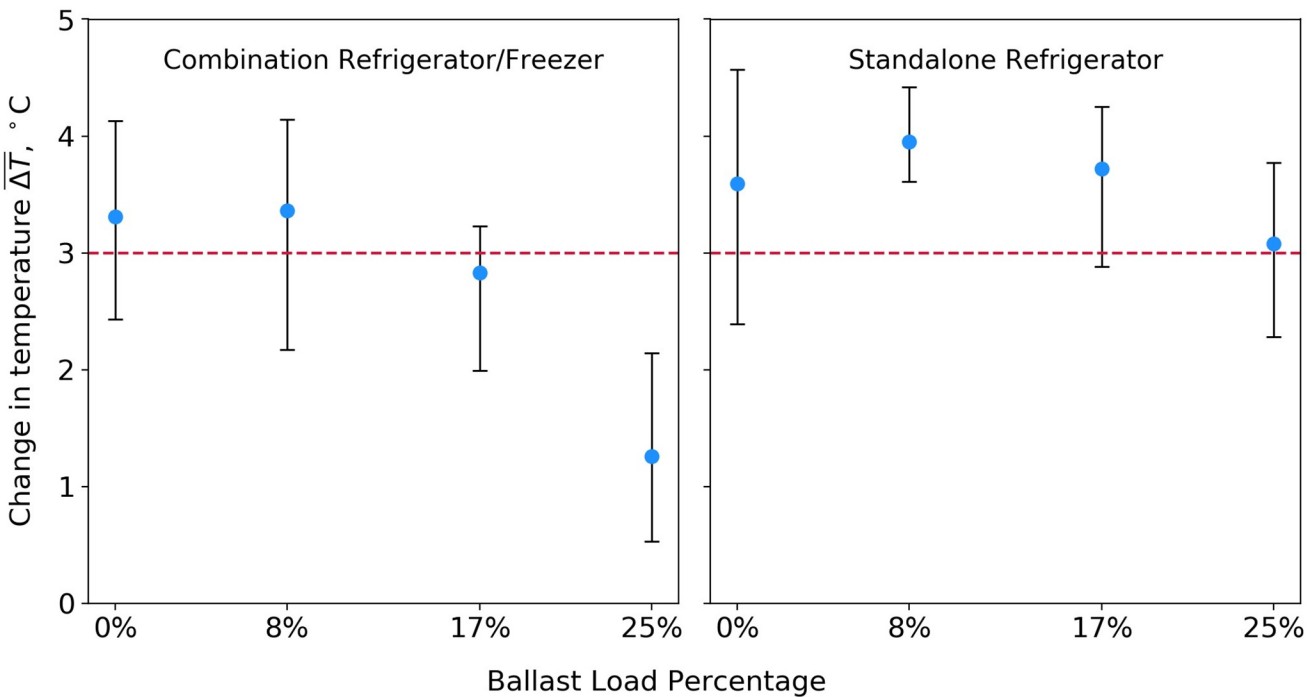

**Fig 5. Refrigerator cabinet temperature changes during repeated door opening trials performed with varying thermal ballast load percentages.** Round blue markers indicate the average temperature rise of all monitored vaccines during each trial, and error bars show the range of these measurements, corresponding to the within-trial variation among monitored vaccines.

## Door open for one hour

Both units were also evaluated with various ballast loads to determine the impact on temperature stability during a worst-case test, in which the refrigerator door was kept open at an angle of 90˚ for one hour. In both units, vaccines were maintained within proper temperature storage conditions for an average of 14 minutes after the door was opened, regardless of the thermal ballast load used (S5 and S6 Tables). The one exception to this occurred during the 25% ballast trial performed in the combination refrigerator. In this case, vaccines remained within a viable storage temperature range for an average of 22 minutes (S5 Table).

## Power outage

Electrical power outages present a significant threat to refrigerated vaccines stored in the absence of a backup power source [22–24]. The U.S. Energy Information Administration (EIA) publishes reliability data provided by domestic utility companies, including frequency and duration of power interruptions experienced per customer each year [25]. As summarized in a recent EIA article [26], U.S. customers experienced, on average, cumulative power interruptions of 7.8 h in 2017, based on the utility-reported System Average Interruption Duration Index (SAIDI). The EIA reports that this figure represents a significant increase from the 2016 SAIDI total of 4.2 h, and may be attributed to the increased incidence of major events, such as hurricanes and winter storms, in 2017. In Maine and Florida, hurricanes and storms resulted in average electrical power interruption durations as long as 42 h in 2017 [26].

The frequency and duration of power disruptions in the United States are projected to increase due to the nation's deteriorating power grid infrastructure [27,28], making power loss mitigation strategies an ever-more critical component of a safe and effective vaccine management program. Prior tests indicated that loading domestic refrigerators with water bottles can extend viable vaccine storage time during power loss situations [6,7], but the precise relationship between thermal ballast quantity and its impact on temperature stability maintenance was not previously evaluated.

The viable vaccine storage times for each of the power loss trials are summarized in Fig 6. A Pearson correlation coefficient, r, was calculated as described in the Evaluation Methods section to assess the strength of the relationship between added thermal ballast and duration of viable vaccine storage time during an outage, for each tested unit. In the standalone refrigerator, $r = 0.96$, $n = 68$, $p < 0.0001$, where n is the total number of observations (temperature sensors multiplied by number of trials), and p is the probability that the null hypothesis (added thermal ballast is not associated with an increase in viable storage time) is true. In the combination unit, $r = 0.94$, $n = 40$, $p < 0.0001$. In both units, we observe a strong positive correlation between increased thermal ballast load and extended viable vaccine storage time during an outage event.

A variety of factors, including refrigerator design, specific heat capacity, total thermal mass and placement of stored items, as well as ambient conditions, are likely to impact the length of time that a unit maintains appropriate vaccine storage temperatures in a power loss situation. Current guidelines stress the importance of continuous temperature monitoring during power outages, and implementation of emergency vaccine storage, handling, and transport procedures, including transfer of vaccines to an alternative storage facility, when warranted [5,29]. However, the findings from our case study may be used to provide some baseline guidance on the performance of vaccine-loaded domestic units during outages of varying lengths. In the absence of a thermal ballast load, vaccines stored in the standalone unit exceeded viable storage temperatures in just over an hour, while vaccines stored in the

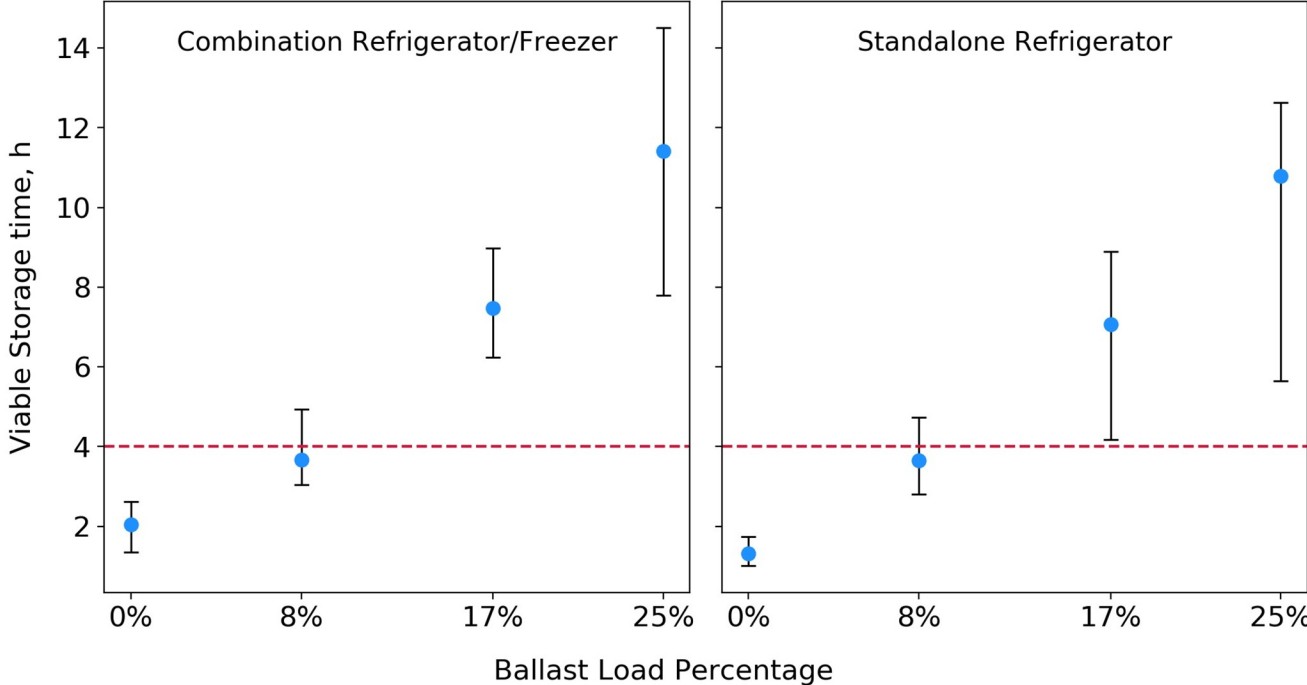

**Fig 6. Viable vaccine storage time during power loss trials conducted with varying thermal ballast load percentages.** Round blue markers indicate the average viable vaccine storage time achieved during each trial, and error bars show the range of these measurements, corresponding to the within-trial variation among monitored vaccines.

combination unit lasted about two hours. In both units, a 17% ballast load extended viable vaccine storage time to more than 6 hours. Under similar operating conditions, a less-cumbersome 10% ballast load can be expected to provide about 4 hours of viable vaccine storage time during an outage.

## Storage containers

The vaccine storage setup used throughout this study incorporated three types of storage containers (stainless steel instrument trays, open plastic shelf bins, and lidded plastic shelf bins). For each of the trials described in the preceding sections, we compared average temperature variations and viable storage time for vaccines stored in each of the different container types. We observed average variations in temperature response of 1 ˚C or less for vaccines stored in different tray types placed on the same refrigerator shelf (S2 Appendix). These variations presented no clear pattern across multiple trials and storage locations, suggesting no clear preference for one type of tray over another.

Numerous thermal engineering studies have evaluated and modeled positional temperature gradients, airflow, and convection patterns in domestic refrigerators [30–34]. It is no surprise that these dynamical variables are the primary drivers of stored product temperatures. As such, the strength of our conclusions about storage container types are limited by our study design, which kept each of the storage container types in fixed positions throughout the study. This configuration makes it impossible to separate the effects of storage location from any effects introduced by storage container type. Still, our results suggest that storage container characteristics (metal, plastic, open vs. closed) have minimal impact on the stability of stored vaccine temperatures.

## Glass shelves vs. wire shelves

During the shelf material case study power outage tests, vaccines experienced too-warm temperature excursions within the first 1.5 hours of the outage period. Vaccines stored in trays placed on glass shelves were maintained within proper storage temperatures ($\overline{\Delta T} < 3°C$) for on average, only 7 minutes longer than identically placed vaccines stored in trays kept on wire shelves. During the power outage recovery phase, vaccines recovered to their pre-outage set-point temperatures within one to one and one-half hours once power was restored. In this case, vaccines in trays stored on glass shelves took, on average, only 3 minutes longer to recover to their pre-outage temperature setpoint, as compared to vaccines in trays kept on wire shelves.

During the 2 h repeated door opening test, approximately 60% of monitored vaccines stored in trays on wire shelves experienced a too-warm temperature excursion, while the remaining 40% were kept within appropriate storage temperatures ($\overline{\Delta T} < 3°C$). By contrast, 100% of monitored vaccines stored in trays placed on glass shelves experienced a too-warm temperature excursion during the test ($\overline{\Delta T} > 3°C$). Vaccines kept on wire shelves also recovered their pre-door opening setpoint temperature faster than vaccines stored on glass shelves (10 min vs. 23 min, on average).

In summary, this limited-scope test demonstrates that vaccines stored on wire shelves in a domestic refrigerator may experience slightly less temperature variability in response to repeated door opening events, as compared to overwise-identically stored vaccines placed on glass shelves. Because this test utilized an after-market glass shelf installation in a domestic refrigerator, these results should not be extrapolated to conclude that glass shelves are universally unfavorable for vaccine storage. Current or future vaccine storage products may exist in which the refrigeration design is successfully engineered to overcome reduced airflow and other challenges introduced by the inclusion of glass shelves.

## Discussion

The results described here support the practice of thermal ballast loading in domestic refrigerators used for vaccine storage. Thermal ballast usage is a practical and effective strategy for mitigating the negative impact of power outage events. Our results indicate that a thermal ballast load comprising 10% to 15% of the total refrigerator storage volume supports sustained vaccine storage temperature maintenance during power loss events for approximately 4 to 6 hours, which is within the range of typical U.S. power outage events. In addition, this window provides ample time for provider action and implementation of remediation plans during extended outages. If no thermal ballast load is used, vaccines stored in domestic refrigerators may experience temperature excursions after just 1.5 hours of sustained outage time.

Thermal ballast usage appears less effective at improving vaccine temperature stability during other types of adverse events. While a small thermal ballast load supports improved temperature stability during defrost cycle activation in a standalone unit, the combination unit maintains appropriate storage conditions during defrost cycle activation regardless of ballast usage.

Thermal ballast usage does not practically or reliably reduce the severity of temperature excursions caused by repeated door opening or accidental "door left open" events. Other tactics, including user training, alarm systems, and built-in engineering controls provide better results in limiting the incidence and negative impact of these types of events [11,12,14,35]. Purpose-built vaccine refrigerators frequently include features like high-powered forced air

convection, "air curtains", and deactivation of fans upon door opening, which limit vaccine exposure to ambient temperatures during routine door openings.

Our findings support the use of a moderate thermal ballast load, comprising 10% to 15% of the total refrigerator storage volume, in domestic units adapted for vaccine storage. Fig 3 shows an example of a 10% ballast load in a standalone refrigerator. While even larger ballast loads could provide additional temperature stability in certain situations, we found loads greater than 17% to be impractical for vaccine storage applications, as too much of the usable refrigerator shelf space is lost to water bottles. Thermal ballast loads less than 10% are also likely to provide some extension in viable vaccine storage time during short power outages, as seen in Fig 6. A moderate ballast load provides an additional programmatic benefit by allowing the refrigerator manager to "cordon off" regions which have been shown to be inappropriate for vaccine storage, including the floor, the top shelf near the cooling vents in a combination unit, and the door of the unit [5–7].

For optimal temperature maintenance and control of stored vaccines, providers should use purpose-built, validated equipment whenever possible. However, in cases where a purpose-built vaccine refrigerator is cost-prohibitive or otherwise unavailable, domestic refrigerators may be used to safely store vaccine with additional training, oversight, and programmatic controls, such as thermal ballast loading.

The small sample size included in this study (two domestic refrigerator units) may limit the generalizability of the findings to some degree. However, because we found that thermal ballast loading is primarily useful as a power loss mitigation strategy, we expect this effect to extend to any refrigerator model. During a power outage, the rate of temperature rise inside a closed refrigerator is driven by the temperature difference between the interior and the environment, the rate of heat exchange through the refrigerator walls, and the heat capacity of the refrigerator's contents. Refrigerator model-specific characteristics like the insulation efficiency will influence the rate of heat exchange. However, the total heat capacity of a refrigerator's contents is a function of those contents' mass and specific heat capacity and is therefore model-independent. Pre-emptively loading a unit with added mass in the form of thermal ballast materials is a practical strategy for extending viable vaccine storage time during power loss events.

While our results indicate that a thermal ballast may be used to improve the temperature stability of vaccines stored in domestic refrigerators subject to certain adverse events, it may also be possible to negatively impact stored vaccines via excessive or improper ballast loading. While we did not identify disruptive temperature gradients created by the thermal ballast loading strategies employed in this study, careless placement of water bottles may unfavorably disrupt refrigerator airflow, exacerbating existing temperature gradients, or creating new ones. Furthermore, most users utilize a single temperature monitoring probe inside their vaccine refrigerators, and so may fail to detect these types of problems [36]. In addition, space constraints imposed by use of even moderate thermal ballast loads could lead to cramped vaccine storage conditions and complicate inventory management.

We emphasize the fact that our study was designed to maintain the same ratio of bottles per location for each ballast load percentage, and as a result, the water bottle placement shown in Fig 3 may not represent the ideal setup for all units. In practice, we suggest filling the door of the unit first, then the floor of the unit, and finally the top shelf, since these areas are typically not suitable for vaccine storage. Users should take care to maintain airflow gaps between each refrigerator shelf and adjacent walls, to avoid creating dead pockets of warmer or colder air.

Finally, the suggestions in this publication are intended for application to domestic food storage refrigerators used to store vaccines and should not be extrapolated to purpose-built vaccine storage refrigerators. Purpose-built refrigerators are typically engineered to maintain appropriate vaccine storage temperatures throughout their entire usable storage volume

during routine use. Most of these units undergo extensive temperature stability and performance validation by the manufacturer. Placement of a large thermal ballast load in a purpose-built unit may disrupt airflow distribution systems, resulting in unfavorable vaccine storage conditions. Users of purpose-built vaccine refrigerators should avoid following the thermal ballast loading practices described in this publication in the absence of manufacturer approval and model-specific guidance.

## Conclusions

We have quantified the effectiveness of thermal ballast loading using filled water bottles, a popular vaccine management strategy, in two domestic refrigerator units subjected to a variety of typical use conditions and adverse events. We observed a strong positive correlation between the thermal ballast load and viable vaccine storage time during power outages (standalone refrigerator, $r = 0.96$, $n = 68$, $p < 0.0001$ and combination refrigerator/freezer, $r = 0.94$, $n = 40$, $p < 0.0001$). Based on our findings, a moderate thermal ballast load, comprising ten to fifteen percent of the total refrigerator storage volume, supports maintenance of vaccine temperatures between 2 ˚C and 8 ˚C for 4 to 6 hours without power, and represents a practical strategy for mitigating temperature excursions risks in areas with frequent or protracted power outages. However, clinics utilizing backup generators and other power loss mitigation strategies have little to gain from this practice, as it does not reliably reduce the frequency or severity of temperature excursions caused by repeated door opening, accidental "door left open" events, or refrigerator defrost cycle activation. Furthermore, while thermal ballast loads greater than 17% provided additional viable vaccine storage time during power loss events, excessive thermal ballast usage restricts available vaccine storage space and could unfavorably disrupt refrigerator airflow. These findings support better, safer vaccine management by enabling providers to make informed, evidence-based decisions about thermal ballast usage.

## Supporting information

**S1 Fig. Photos of 25% and 8% thermal ballast load setups.**
(DOCX)

**S2 Fig. Temperature response of seventeen monitored vaccine vials to defrost cycle activation in standalone refrigerator, with 0% thermal ballast load.**
(PDF)

**S1 Table. Combination refrigerator/freezer defrost cycle.** $\overline{\Delta T}$ data shown in Fig 4L. "Pos error" and "neg error" show the extreme values recorded by all sensors, corresponding to the overall range.
(CSV)

**S2 Table. Standalone refrigerator defrost cycle.** $\overline{\Delta T}$ data shown in Fig 4R. "Pos error" and "neg error" show the extreme values recorded by all sensors, corresponding to the overall range.
(CSV)

**S3 Table. Combination refrigerator/freezer repeated door opening.** $\overline{\Delta T}$ data shown in Fig 5L. "Pos error" and "neg error" show the extreme values recorded by all sensors, corresponding to the overall range.
(CSV)

**S4 Table. Standalone refrigerator repeated door opening.** $\overline{\Delta T}$ data shown in Fig 5R. "Pos error" and "neg error" show the extreme values recorded by all sensors, corresponding to the overall range.
(CSV)

**S5 Table. Combination refrigerator/freezer door left open.** Average duration of viable vaccine storage time with refrigerator door left open.
(CSV)

**S6 Table. Standalone refrigerator door left open.** Average duration of viable vaccine storage time with refrigerator door left open.
(CSV)

**S7 Table. Combination refrigerator/freezer power outage.** Average duration of viable vaccine storage time, as shown in Fig 6L. "Pos error" and "neg error" show the extreme values recorded by all sensors, corresponding to the overall range.
(CSV)

**S8 Table. Standalone refrigerator power outage.** Average duration of viable vaccine storage time, as shown in Fig 6R. "Pos error" and "neg error" show the extreme values recorded by all sensors, corresponding to the overall range.
(CSV)

**S1 Appendix. Comparison of temperature response by monitored vaccines stored in different types of trays.**
(PDF)

**S2 Appendix. Comparison of door opening temperature response by boxed and unboxed vaccine, at 0% and 25% thermal ballast loads.**
(PDF)

## Acknowledgments

The authors thank W. Wyatt Miller for technical assistance and fabrication support in this study.

## Author Contributions

**Conceptualization:** Michal Chojnacky.

**Formal analysis:** Michal Chojnacky, Alexandra L. Rodriguez.

**Funding acquisition:** Michal Chojnacky.

**Investigation:** Alexandra L. Rodriguez.

**Methodology:** Michal Chojnacky.

**Validation:** Alexandra L. Rodriguez.

**Visualization:** Michal Chojnacky.

**Writing – original draft:** Michal Chojnacky, Alexandra L. Rodriguez.

**Writing – review & editing:** Michal Chojnacky.

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
