## [Decision Letter · Decision Letter 0]

10 Mar 2020

PONE-D-20-03640

Effect of thermal ballast loading on temperature stability of domestic refrigerators used for vaccine storage

PLOS ONE

Dear Chojnacky,

Thank you for submitting your manuscript to PLOS ONE. After careful consideration, we feel that it has merit but does not fully meet PLOS ONE’s publication criteria as it currently stands. Therefore, we invite you to submit a revised version of the manuscript that addresses the points raised during the review process.

We would appreciate receiving your revised manuscript by Apr 24 2020 11:59PM. To enhance the reproducibility of your results, we recommend that if applicable you deposit your laboratory protocols in protocols.io, where a protocol can be assigned its own identifier (DOI) such that it can be cited independently in the future. For instructions see: http://journals.plos.org/plosone/s/submission-guidelines#loc-laboratory-protocols

We look forward to receiving your revised manuscript.

Kind regards,

Dai-Viet N. Vo, Ph.D.

Academic Editor

PLOS ONE

Journal Requirements:

"I have read the journal's policy and the authors of this manuscript have the following competing interests:

This study was partially funded by an Interagency Agreement with Centers for Disease Control and Prevention (CDC) and the authors' institution. As part of this agreement, MC provides technical guidance and research support to the CDC's Vaccines for Children program. Additionally, MC is a member of the NSF Joint Committee for Vaccine Storage, a consensus body responsible for creating and maintaining an ANSI standard for purpose-built vaccine storage equipment.".

i) Please confirm that this does not alter your adherence to all PLOS ONE policies on sharing data and materials, by including the following statement: "This does not alter our adherence to  PLOS ONE policies on sharing data and materials.” (as detailed online in our guide for authors http://journals.plos.org/plosone/s/competing-interests).  If there are restrictions on sharing of data and/or materials, please state these. Please note that we cannot proceed with consideration of your article until this information has been declared.

ii) Please include your updated Competing Interests statement in your cover letter; we will change the online submission form on your behalf.

Reviewers' comments:

Reviewer's Responses to Questions

**Comments to the Author**

1. Is the manuscript technically sound, and do the data support the conclusions?

Reviewer #1: Yes

Reviewer #2: Yes

2. Has the statistical analysis been performed appropriately and rigorously? 

Reviewer #1: Yes

Reviewer #2: Yes

3. Have the authors made all data underlying the findings in their manuscript fully available?

Reviewer #1: Yes

Reviewer #2: Yes

4. Is the manuscript presented in an intelligible fashion and written in standard English?

Reviewer #1: Yes

Reviewer #2: Yes

5. Review Comments to the Author

Reviewer #1: The study entitled “Effect of thermal ballast loading on temperature stability of domestic refrigerators used for vaccine storage” by Michal Chojnacky et al. investigated the effect of thermal ballast loadings on the temperature stability of vaccine stored. In general, the manuscript is well written and organized, the language is excellent so it is easy to follow. The study is every interesting because the authors compared their results on two kinds of refrigerators involving combination and standalone. Moreover, many factors (defrost cycle, door open, power outage, storage containers, glass shelves vs. wire shelves) have been studied carefully. The results seem reliable with repeatable runs. The prospective of this study is promising and meaningful practically since it can be applied for isolated regions with a deficiency of cooling instruments or electricity. With these reasons, the reviewer is in favor of acceptance for publication after the manuscript is revised along with the following minor comments.

1. The abstract of this manuscript seems quiet long and it should also be appropriately adhered according to the journal’s guide, please check and shorten it. By the way, please check the table of reference contents.

2. In the conclusion, I think that you should correct first two sentences because they don’t come from your results. The conclusion should be a summary in the method, the most highlighted results, what the findings attained, etc. Please insert some numeric data to more consolidate the significance of this study.

3. In Figs. 1-3, the name and position of the important materials (thermos ballast, vaccines, etc.) should be clarified. In addition, each type of vaccine is characterized by various conditions in critical temperature, light-exposure intensity, expiration date, etc. The same case is also applied for refrigerator. Please list these properties of those vaccines/refrigerator in your experiments to make sure that the further studies can be repeated to verify.

4. In Fig. 6, there are increasing trends in viable storage time along with the percentage of ballast loading for both combination and standalone refrigerators. You are highly recommended to increase ballast loadings (e.g. 30%, 35%, 40%, or more) to track the result and detect the optimum ballast loadings.

Reviewer #2: This work investigated the effect of thermal ballast loading on temperature stability of domestic refrigerators used for vaccine storage. The work is interesting with relevant application to the healthcare area. The discussion is well-written; nonetheless, certain parts especially the method section are unclear. Therefore, I would like to suggest a minor revision for this work.

1) Introduction

a) Please specify what kind of “certain” operating conditions.

“However, under certain operating conditions, domestic refrigerators can expose vaccines to temperatures outside the permissible storage range (8–13).”

2) Refrigerator Selection

a) How do the authors confirm the variation of shelve type do not affect the performance of refrigerator?

“The combination unit had three glass storage shelves inside its refrigerated compartment, and the standalone unit had four interior wire shelves.”

b) More specifications of refrigerators should be provided in Table 1. For instance, power consumption and etc.

c) The model of refrigerators should be mentioned.

3) Temperature Monitoring

a) Any specific reason for using different number of thermocouples to monitor vaccines from different refrigerators?

“A total of twenty-nine Type T thermocouples were used to monitor vaccines in the two refrigerators, with twelve monitored vaccines in the combination unit, and seventeen monitored vaccines in the standalone unit.”

b) What is type A and type B temperature measurement systems? Are them refers to thermocouple? If yes, why use two different thermocouples? Confusing as they are not mentioned in the text.

4) Thermal Ballast

a) Are the large bottles and jugs also made from plastic? Different matters might have different heat conductivity.

b) The formula of temperature stability should be included in the methodology section instead of results & discussion section.

5) Test Protocol

a) The set temperature should be specified in the text.

6) Defrost cycle

a) Transient profile of temperature should be provided to indicate any fluctuation over time.

b) Please specify how many % of reduced temperature fluctuation inside bracket.

“…while the 8 % and 17 % ballast loads had little effect on the defrost cycle temperature response.”

7) Repeated Door Opening

a) Redundant discussion. The way of evaluation should be declared in methodology rather than discussion.

“…with Δ > 3 ° indicative of a potential excursion condition.”

8) Door Open for One Hour

a) Similar to comment #6a.

“As before, we examined the average temperature change of the monitored vaccine vials, Δ, to assess temperature stability during the trial. We define “viable vaccine storage time” as the duration of time in which vaccines were maintained within 3 °C of their initial, pre-door opening storage temperature (Δ < 3 °).”

b) I suggest the authors to provide the data as supplementary data.

“In this case, vaccines remained within a viable storage temperature range for an average of 22 minutes (data not shown).”

9) Power Outage

a) Similar to comment #6a.

“To determine the impact of thermal ballast load on refrigerated vaccine temperature stability during a power outage event, we examined “viable vaccine storage time” during simulated outage tests performed at different ballast load percentages. As before, viable vaccine storage time is defined as the duration of time during which vaccine temperatures were maintained within 3 °C of their initial, pre-outage storage temperature (Δ < 3 °). In this framework, a temperature rise of more than 3 °C is equivalent to the start of a temperature excursion, and the end of viable vaccine storage time.”

b) How the Pearson correlation coefficient being calculated? It should be included under the methodology section.

“Pearson correlation coefficient, r, was calculated to assess the strength of the relationship between added thermal ballast and duration of viable vaccine storage time during an outage, for each tested unit.”

“…n is the total number of observations (temperature sensors multiplied by number of trials), and p is the probability that the null hypothesis (added thermal ballast is not associated with an increase in viable storage time) is true”

10) Storage Containers

a) How trivial it is? Please provide some numerical values or graph?

“Still, our results suggest that the selection of a storage container (metal, plastic, open vs. closed) is trivial in terms of its impact on maintaining stored vaccine temperatures.”

11) Glass Shelves vs. Wire Shelves

a) The methodology of this section should be included under the methodology section.

b) Please provide the specification of the glass sheet that used to produce glass shelves.

12) Discussions

a) It is well-written. Well done.

13) Conclusion

a) The conclusion should be more specific in accordance to the current findings. Please support your recommendation/suggestion with facts like data.

6. PLOS authors have the option to publish the peer review history of their article (what does this mean?). If published, this will include your full peer review and any attached files.

Reviewer #1: No

Reviewer #2: No

---

## [Author Response · Author response to Decision Letter 0]

30 May 2020

Response to Reviewers letter has been uploaded as a separate file.

---

## [Decision Letter · Decision Letter 1]

23 Jun 2020

Effect of thermal ballast loading on temperature stability of domestic refrigerators used for vaccine storage

PONE-D-20-03640R1

Dear Dr. Chojnacky,

We’re pleased to inform you that your manuscript has been judged scientifically suitable for publication and will be formally accepted for publication once it meets all outstanding technical requirements.

Kind regards,

Dai-Viet N. Vo, Ph.D.

Academic Editor

PLOS ONE

Additional Editor Comments (optional):

The paper has been properly revised. Thus, it could be considered for publication.

Reviewers' comments:

Reviewer's Responses to Questions

**Comments to the Author**

1. If the authors have adequately addressed your comments raised in a previous round of review and you feel that this manuscript is now acceptable for publication, you may indicate that here to bypass the “Comments to the Author” section, enter your conflict of interest statement in the “Confidential to Editor” section, and submit your "Accept" recommendation.

Reviewer #1: All comments have been addressed

Reviewer #2: All comments have been addressed

2. Is the manuscript technically sound, and do the data support the conclusions?

Reviewer #1: Yes

Reviewer #2: Yes

3. Has the statistical analysis been performed appropriately and rigorously? 

Reviewer #1: Yes

Reviewer #2: Yes

4. Have the authors made all data underlying the findings in their manuscript fully available?

Reviewer #1: Yes

Reviewer #2: Yes

5. Is the manuscript presented in an intelligible fashion and written in standard English?

Reviewer #1: Yes

Reviewer #2: Yes

6. Review Comments to the Author

Reviewer #1: I suggest to accept this manuscript for publication because the authors have responded all of the comments correctly and appropriately.

Reviewer #2: The manuscript reported an interesting work on "Effect of thermal ballast loading on temperature stability of domestic refrigerators used for vaccine storage". Well done for the improvement. In my opinion, the manuscript is ready for publication. All the comments have been properly addressed.

7. PLOS authors have the option to publish the peer review history of their article (what does this mean?). If published, this will include your full peer review and any attached files.

Reviewer #1: No

Reviewer #2: No

---

## [Editor Report · Acceptance letter]

26 Jun 2020

PONE-D-20-03640R1 

Effect of thermal ballast loading on temperature stability of domestic refrigerators used for vaccine storage 

Dear Dr. Chojnacky:

I'm pleased to inform you that your manuscript has been deemed suitable for publication in PLOS ONE. Congratulations! Your manuscript is now with our production department. 

Kind regards, 

on behalf of

Dr. Dai-Viet N. Vo 

Academic Editor

PLOS ONE